# Sustainable Process and Product Innovation in the Eyewear Sector: The Role of Industry 4.0 Enabling Technologies

Federica Murmura [1],*, Laura Bravi [1] and Gilberto Santos [2]

[1] Department of Economics, Society, Politics, University of Urbino Carlo Bo, Via Saffi 42, 61029 Urbino, Italy; laura.bravi@uniurb.it
[2] School of Design, Polyt. Institute Cavado Ave, Campos do IPCA, Rua do Aldão, 4750-810 Barcelos, Portugal; gsantos@ipca.pt
* Correspondence: federica.murmura@uniurb.it

**Abstract:** This paper aims to provide the reader with an organic view of the eyewear sector considering both market and quality aspects and evaluating the role of Industry 4.0 in process and product innovation for managing consumer health, analyzing a case study of a leading multinational company in the eyewear and ophthalmic lenses sector. The research has been developed with a qualitative approach. The study is a conceptual development and it uses an exploratory interview to create a single case study. The case study was developed with the realization by the researcher of a semi-structured interview. The selected interlocutor was the Innovation Manager of Alpha Optics. It has been decided to focus the attention on this figure, as it was responsible for the realization and introduction into the company of Industry 4.0 enabling technologies for developing health innovations. From this case study it was possible to observe how the connection with the trends that influence the demand for eyeglasses is a driving factor for product innovation. Products increasingly adapted to the needs of young people and the use of digital devices seem to be the ones on which the greatest number of innovations are concentrated.

**Keywords:** industry 4.0; enabling technologies; eyewear sector; case study; innovation

## 1. Introduction

The constant technological innovations, their rapid diffusion and application in various fields are radically changing the global economic scenario. Scholars have long debated (and still debate) the possibility of understanding these new changes as the result of the advent of a new industrial revolution, failing, however, to agree on the name of this revolution. Some theorists, in fact, argue that the digitalization of the manufacturing system represents the third industrial revolution, others, on the other hand, claim that changes in manufacturing have gone further and that for speed and impact they represent a new revolution, contributing to the birth of a fourth industrial revolution [1]. National governments embraced this last perspective and have taken steps to launch special plans that could support the so-called Industry 4.0 [2]. This term indicates a process generated by technological transformations in the design, production and distribution of manufacturing systems and products, aimed at automated and interconnected industrial production. In particular, it identifies an organization based on the digitization of all stages of production processes. From this definition it can be understood how the physiognomy of companies is changing considerably, gradually abandoning their Fordist character, and moving towards a new business configuration: that of a Smart Industry [3].

In a constantly evolving sector such as eyewear, Industry 4.0 has a significant impact on the innovation of machinery, for the production of the finished product or in those used in intermediate processes, and of products, with innovations on ophthalmic lenses and frames (both for prescription and for sunglasses) [4].

When defining the eyewear market, one of the first difficulties lies in circumscribing its perimeter. Very often, in fact, by eyewear it means everything that has to do with the design, production and sale of optical frames and sunglasses. However, following this reasoning, it can be said that an equally important category such as that of ophthalmic lens manufacturers would be excluded from this definition. In fact, it is undisputed to argue that an optical frame, alone, cannot correct vision defects without lenses, just as sunglasses cannot protect from the sun's rays without the appropriate filters. Extending the reflection to lens manufacturers also becomes necessary when the boundary between these two sectors becomes more blurred, as taught by the fusion of great giants of the market, of frames and ophthalmic lenses [5].

The product innovations analyzed in the category of frames can be traced back to three main themes: the first, inherent to the growing interest of consumers, and therefore of companies, in ecological issues; the second, linked to the transposition of tailoring concepts into eyewear, giving life to the manufacture of totally tailor-made eyewear; the third, concerning smart glasses, that is the tendency to make glasses an extension of the devices we use every day or to expand their application scope to other purposes beyond the correction and/or protection of eyesight [6]. The use of these hybrid devices between sunglasses or eyeglasses and smart accessories in certain business contexts (such as the construction sector, the automotive sector, the transport sector, the utility industry, etc.) seems to be destined to grow: it is estimated that by 2025 the global smart glasses market will reach 20.1 million dollars with an annual growth rate of 16.2. The 79% of companies that have adopted wearable technology say it is a strategic factor for the future of their company [7]. As for the innovations in the sector of optical frames and sunglasses, there is also a high rate of innovation in the lens sector; among this is the creation of eco-friendly lenses emerges. They are called Bio Lens 2.0 and are 100% biodegradable lenses obtained from a modified polymer that drastically reduces biodegradation times. The Bio Lens 2.0 can be used both in the creation of demo lenses, that is the non-graduated ones that accompany the optical frame in the phase of choice of the same and that therefore are the ones discarded most frequently, and sun lenses [8]. A first process innovation could be the one introduced by Kering in the management of its outbound logistics. The new big company in the luxury eyewear sector has introduced a recent highly automated system, with which it is possible to control every aspect of the supply chain. Furthermore, through the total centralization of stocks, Kering aims to reduce the stagnation of products, significantly reducing warehouse costs. To date, Kering can count on 5 million pieces stored, 35,000 pieces handled per day and four daily shipments, a system that has made the company among the best in class in the distribution process [9]. Among the best in class can also be found Luxottica, with a distribution system serving both the company's businesses (both wholesale and retail) and integrated internationally thanks to the use of a centralized production platform. The latter allows, in fact, the constant monitoring of the balance between sales and stock levels, also making use of the strategic positioning of its distribution centers to target markets [10–12]. Process innovations can also be considered those that can be found much further down the value chain; it is the process of fitting prescription lenses into a pair of glasses. Both retailers and specialized laboratories today can count on the use of innovative machinery that allows the reduction in the number of steps involved in the process of shaping a lens (based on the shape of the wearer's glasses) and its mounting on the frame.

This paper aims to provide the reader with an organic view of the eyewear market considering both market and quality aspects in the eyewear sector, evaluating the role of Industry 4.0 in process and product innovation for managing consumer health, analyzing a case study of a leading multinational company in the eyewear and ophthalmic lenses sector, that for privacy reasons will be called Alpha Optics. Therefore, the research question that the paper investigates is the following:

*RQ1: How the enabling technologies of Industry 4.0 can be used to introduce elements of quality and product and process innovation in the eyewear sector?*

The paper is organized as follows: Section 2 defines the methodology used for the development of the case study, Section 3 presents the main results and discusses them and finally Section 4 draws the main conclusions.

## 2. Theoretical Background

Advances in communications technology, sensing devices and big data processing techniques have driven the expansion of the Internet of Things (IoT) into smart home applications, environmental monitoring, healthcare, inventory management, public security, smart transport and smart logistics [13,14].

It is common to use the term Industry 4.0 to refer to this new industrial revolution, which today is seen as the process of digitizing the manufacturing sector that, by renewing the value chain, changes the way of working but also the nature of organizations [15]. The level of innovation is such that today the synonym of Industry 4.0 is smart manufacturing, where the suffix "smart" becomes the common denominator of an integrated management of information, associated with the use of digital technology [16].

Although there is no commonly accepted definition, Industry 4.0 is generally seen as a process that will culminate in a new conception of industry, from the development of new products and services, to research and innovation, to validation and production, with the lowest common denominator being a high degree of automation and interconnection [17].

What is more, in support of manufacturing enterprises, Industry 4.0 technologies have made production processes more efficient and less impactful [18,19]. Particularly, among these disruptive technologies, Additive Manufacturing (AM) better known as 3D printing, has attracted attention as a 'game-changer' for conventional manufacturing regarding opportunities for mass customization, local (distributed) production, no-waste printing process, spare part printing for repair and remanufacturing [20,21].

In addition, the new Industry 4.0 management and data collection tools, which are able to collect and timely process data, can facilitate companies in the evaluation of the sustainability practices introduced [22]. The implementation of Industry 4.0 technologies, other than generating numerous benefits for companies, requires them to transform their value proposition through a business model innovation process [18,23], that can be of a limited entity if the introduction of new technologies consists of incremental innovation, while in the case of radical innovations, the process can lead to a total reconfiguration of the company's value proposition [24]. The change in business model that is taking place consists of considering the relationship between innovation and sustainability, defining sustainable and smart products (SSP), that is a new generation of smart products to achieve circular economy and sustainability. SSP is characterized by servitization, sustainability objectives and environmental care [25,26]. In this context, it is increasingly important for organizations to adjust the innovation strategy for SSP development by implementing a sustainable business model to reduce their environmental and social impacts [27]. One framework supporting this focus is the framework of dedicated innovation systems (DIS), that "explicitly go beyond technological innovation and economic growth and allow for paradigmatic change towards sustainability: They are "dedicated" to foster the joint search for transformative innovations. In other words, the proposed conception of DIS implies that the predominant focus of innovation systems on economic competitiveness needs to step back behind the global societies' imperative of sustainability" [28,29]. In this context it is important to consider new roles of government in motivating open innovation dynamics from the perspective of system dynamics [30]. However, as stated by Lee et al. [31] it is important to point out that the waves of innovation that the new paradigm will bring entail the development, deployment, and exploitation of technologies, but their initiation and growth are strongly affected by the creative characteristics of organizations.

The features just described are the basis for the design and development of innovative systems in different application fields. In particular, in the ophthalmology and eyewear sector, the possibility of guaranteeing the originality and certification of each product, considering also the need to guarantee a wide product variability, is of great importance [5,14].

As stated by Sicari et al. [5] in this case, the eyewear field is the ideal candidate to become the smart object. Every year different brands put on the market products that have unique technical features and design aspects. From this it derives the importance of providing robust solutions for guaranteeing the originality and healthiness of the products in the eyewear sector since consumers must be sure that the products they buy do not harm their health.

However, in the last ten years the eyewear sector has been very little taken into consideration in the literature and it has been connected to the Industry 4.0 only with regard to the theme of augmented reality and smart glasses [14,32]. There are no works that study in depth how the sector itself is innovating its products and processes in a sustainable and competitive way thanks to the use of the enabling technologies of Industry 4.0.

For this reason, this study aims to try to fill this gap, considering how and with what level of depth the enabling technologies of Industry 4.0 can be applied to this sector, to obtain relevant product and process innovations. To do this, it was decided to take into consideration the Alpha Optics case study, which allows to understand in depth how these technologies have been implemented in a company that achieved a good position in the market, where there were already many competitors thanks to innovations.

## 3. Materials and Methods

The research has been developed with a qualitative approach. The study is a conceptual development and it uses an exploratory interview to create a single case study of Alpha Optics. The company name is fictional, for privacy reasons. The case study allowed for examining in depth the role of Industry 4.0 technologies in product and process innovations, considering among the identified enabling technologies, which and how many are used by a company active in the eyewear sector, what role they play in product and processes innovation and how they change the physiognomy of a company [33].

The case study was developed with the realization by the researcher of a semi-structured interview. It was considered to be the methodology that best suited the needs of the research, as it allows the interviewer to decide, in relation to the progress of the interview, how to articulate the topics in order to obtain more useful information on the salient topics and insights where it was deemed appropriate [34]. The comparative advantages of case study methods include identifying new or omitted variables and hypotheses, examining intervening variables in individual cases to make inferences on which causal mechanisms may have been at work, developing historical explanations of particular cases, attaining high levels of construct validity, and using contingent generalizations to model complex relationships such as path dependency and multiple interactions effects. Particularly important is the ability to identify new hypotheses, which case studies can do through a combination of deduction and induction [35]. Most case study researchers have argued that single-case studies can provide tests that might strongly support or impugn theories [36]. Of course, qualitative studies have their own limitations [37], which has been also discussed in the conclusion section of this paper. Case study findings are usually contingent and can be generalized beyond the type of case studied only under specified conditions, such as when a case study shows that a variable is not a necessary condition or a sufficient condition for an outcome, or when a theory fails to fit a case that it appeared most likely to explain. Potential limitations of case studies, though not inherent in every one, include indeterminacy or the inability to exclude all but one explanation, lack of independence of cases, and the impossibility of perfectly controlling case comparisons [35,38].

The selected interlocutor was the Innovation Manager of Alpha Optics. It has been decided to focus on this company since it was found to be one of the companies most committed to product innovation not only in its field of activity, ophthalmic lenses, but also in areas other than its business such as that of frames with the development of a project that deeply exploited one of the enabling technologies of Industry 4.0, that is additive manufacturing. Moreover, it has been decided to focus the attention on the Innovation

Manager of the company, as it was the figure responsible for the realization and introduction into the company of Industry 4.0 enabling technologies for developing health innovations. The interview was carried out in September 2020.

At the opening of the interview, consent was requested to start recording the same and the interviewee was provided with general information on the purpose and on the thematic areas covered during the interview (general questions about the company and the interviewee; general character on the concept of innovation in the company; questions on product innovations; questions on process innovations; general questions on total quality management issues).

In addition, at the beginning and during the latter, appropriate terminological clarifications on the concepts of process and product innovation were provided to allow the interviewee to understand and better respond to the questions.

## 4. Results and Discussion

### 4.1. The International and Italian Eyewear Market

The eyewear market is more complex than it appears. In fact, it is possible to segment the entire industry by product, thus obtaining four categories: prescription frames (without lenses), sunglasses, ophthalmic lenses, contact lenses. With the sole exception of contact lenses, not considered in this discussion, Table 1 shows the revenues (Million euros) and sales volumes (Million units) in the world for the three-year period 2017–2019. The compound annual growth rate (CAGR) for total revenues for the years 2017 to 2019 stood at 2.4%.

**Table 1.** World revenues and sales volumes of the eyewear sector for the three-year period 2017–2019.

|  | Revenues (Million Euros) | | | | Volumes (Million Units) | | | |
|---|---|---|---|---|---|---|---|---|
|  | **2017** | **2018** | **2019** | **Tot.** | **2017** | **2018** | **2019** | **Tot.** |
| Sunglasses | 18.253 | 18.809 | 19.344 | 55.806 | 837.9 | 852.5 | 866.8 | 2557.2 |
| Optical frames | 30.577 | 31.347 | 32.105 | 94.029 | 577.0 | 587.1 | 597.0 | 1761.1 |
| Lenses | 46.282 | 47.313 | 48.313 | 141.908 | 684.2 | 693.9 | 703.5 | 2081.6 |
| **Tot.** | 95.112 | 97.469 | 99.764 |  | 2.0991 | 2.1335 | 2.1673 |  |

The sunglasses segment recorded the greatest growth, with an annual growth rate of 2.9% for the same years, although the lens segment led the total revenue for the three-year period for 49%. with 141,908 million euros. Revenues relating to lenses can be explained by the higher price per unit of the same compared to those of optical frames and sunglasses. In fact, in 2019, compared to 703.5 million units of lenses, 163.3 million less than the 866.8 units of sunglasses, ophthalmic lenses have a price per unit of 79.52 compared to 25.64 euros for a pair of sunglasses. Interesting data is that concerning the percentage of sales for online and offline channels: in 2019 88% of purchases were made offline compared to 12% of those online. The numbers of the online channel, however, are destined to grow at the expense of offline channels, which will still retain their leadership. It is estimated that in 2023 the percentage of online purchases will be 21% compared to 79% of those made offline [39].

If the global market numbers seem to be registering constant growth, the same cannot be said for the Italian market. A study conducted by the National Association of Optical Goods Manufacturers (ANFAO) and Confindustria Moda [40] highlighted that 2018 was a difficult year for Italian eyewear. The domestic market, in fact, recorded a further contraction compared to 2017, closing 2018 with a decrease of 0.8% in the overall value. As shown in Table 2, the data relating to both prescription and sun frames show a contraction of the sell-in of 0.7% and 2.3%, respectively, with a total decrease of 1.7%.

**Table 2.** Italian market: sell-in and sell-out at 2018.

|  | Sell-In | | Sell-Out | |
|---|---|---|---|---|
|  | Mln € | % Var. to 2017 | Mln € | % Var. to 2017 |
| Sunglasses | 353,69 | −2,3 | 570 | −4.9 |
| Optical frames | 225,36 | −0.7 | 539 | −4.6 |
| Lenses | 263,35 | +1.4 | 1307 | +2.5 |

Even the data relating to the sell-out, and therefore to consumption, recorded a contraction, closing 2018 with −0.7%. To cushion this figure are undoubtedly ophthalmic lenses, which recorded an increase of 2.5%, together with the competition, albeit marginal, of contact lenses (+0.4%). Ophthalmic lenses, in fact, represent 45.1% of the eyewear turnover. As for the points of sale on the national market, there is a supremacy of independent points of sale or that are part of large groups compared to chains: of 9310 points of sale scattered throughout the country, 91% are independent points of sale. On the other hand, the number of companies present on the national territory remained almost unchanged: 867 in 2018, only four more than in 2017. Thanks to the contribution of the large multinationals present in the industry, the numbers relating to employees in the sector recorded an increase of 2.3% compared to 2017.

As regards import-export for Italy (Table 3), in 2018 there were no contractions even if the overall performance could be considered mediocre. In fact, exports of prescription frames, sunglasses and lenses have increased by 0.2% compared to the previous year, for a total of 3738 million euros. Driving the export numbers, once again, are ophthalmic lenses, with an increase of 8.4%, followed by sunglasses (+1.2%) and prescription frames (+0.4%). The geographic areas where it exports most are Europe (49.7%), America (32.3%) and Asia (16%) followed by Africa (1.5%) and Oceania (0.5%). In these last three continents there was a contraction in the export of optical frames and sunglasses of 2.2%, 16.2% and 19%, respectively. America and Europe, on the other hand, recorded an increase of 2.2% and 2% driven, respectively, by good performances in North America and Mexico and by the increase in exports to the countries of Northern and Eastern Europe. 2018 ended with a total of 1270 million euros: 1.9% more than the previous year.

**Table 3.** Italian eyewear market: main data.

|  | Revenue | | Production | | Domestic Market | | Import | | Export | |
|---|---|---|---|---|---|---|---|---|---|---|
|  | Mln € | % Var. to 2017 | Mln € | % Var. to 2017 | Mln € | % Var. to 2017 | Mln € | % Var. to 2017 | Mln € | % Var. to 2017 |
| **2018** | 5095 | 0.7% | 3865 | 1.6% | 987 | −0.8% | 1270 | 1.9% | 3738 | 0.2% |

### 4.2. Alpha Optics and Its Innovative Health Profile in the Eyewear Sector

One of the first issues addressed during the interview was the history, activities and functions of the Italian branch of Alpha Optics today.

Alpha Optics, in fact, took its first steps as a commercial office in Milan, in 1982, whose functions consisted substantially of receiving and shipping orders to its customers. A first step towards today's configuration of the branch was taken in 1989, the year in which a production laboratory was inaugurated in Cinisello Balsamo in which, starting from a semi-finished product, Alpha Optics was able to produce lenses and apply surface treatments to the lenses on the Italian territory.

A further leap in the size of the company has taken place in the years between 1989 and 1998 when, due to the increase in sales and distribution, Alpha Optics decides to equip itself with a new plant in Garbagnate.

[...] a single site where to have all the departments together [...] built according to what were the optimal layouts for Alpha Optics ", which owned" [...] in the world, a whole series of construction workshops with its own standard layout, optimized over time, based on experience and new technologies.

Today Alpha Optics Italy serves the domestic and Maltese markets, with a turnover that in the last fiscal year exceeded 50 million euros. The current plant of the company can count, in addition to the main office in Garbagnate, on a further four branches (Rome, Bari, Palermo and Catania) that deal with the distribution and provision of services. The activities involved in the main office are:

- activities related to the production process: production of lenses, treatments, eyewear assembly department and remote shaping department;
- activities related to logistics: a single warehouse for semi-finished products for the production of lenses, a main warehouse with ready-made or semi-ready products, that is "a middle ground between the semi-finished product and a ready lens";
- marketing activities;
- administrative activities;
- sales administration activities;
- customer service activities;
- human resource management activities;
- service activity: checking of return/return lenses;
- production management activities

Of these activities, only three are replicated in the other branches throughout the country (customer service, warehouse, eyewear assembly).

By transposing the activities listed by the Innovation Manager into Porter's value chain, it can be seen how Alpha Optics seems to own all the primary and support activities of the Porter chain, with the exception of the research and development activities which, it can be deduced, are conducted in the parent company, with some small exceptions [41]. The representation of the primary and secondary processes and the phases involved in the production process of Alpha Optics eyewear are represented in Figure 1. Three macro phases that contribute to the development and production of eyewear can be identified, as also suggested by De Toni e Nassimbeni [42]:

- creative phase: stylists and designers formalize the aesthetic content of the product, which varies in shapes and colors in line with seasonal fashion variations.
- Design phase (technical): the product concept is converted into a project containing the technical specifications and production tools.
- Production phase: after being tested, the tools necessary to create the new model are produced.

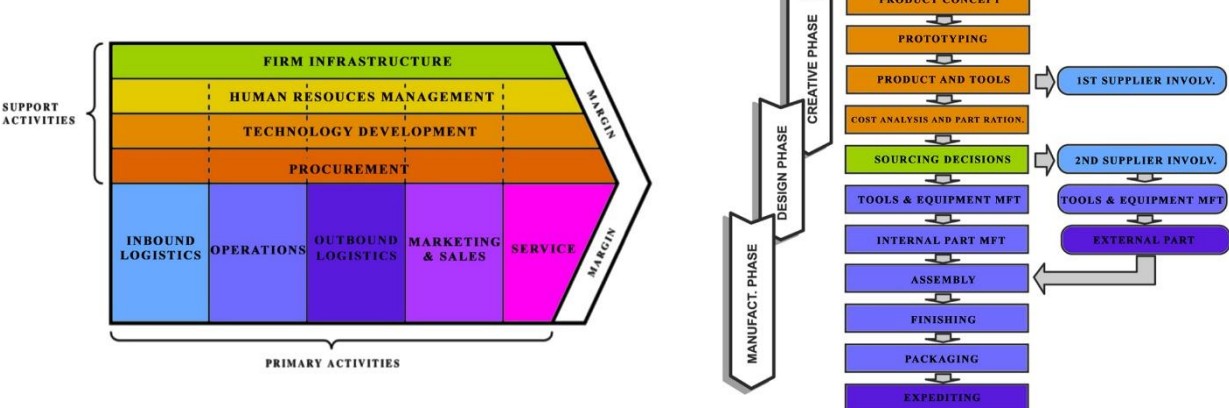

**Figure 1.** Representation of business processes and points of contact between the value chain and product development activities.

Each macro phase, in addition to being strictly interconnected with the others, can be broken down into further steps. After appropriate market research, the creative phase begins with the product concept, which formalizes the initial idea of the new model. The prototyping phase follows with the construction of a tangible model of the eyewear as it was conceived in the previous phase. The prototype allows the verification of the presence of any gaps between the conception and the actual reproducibility of the same: in fact, the technical specifications of the glasses are identified here as well as any difficulties encountered in production. At this point, everything necessary for the mass production of the prototype (product and tools) can be designed, which is broken down into its various components (rods, nose pads, bridges, terminals, hinges, screws, etc.) so as to be able to determine the specification techniques and the design of each.

It is in this phase that the involvement of external suppliers is identified which, according to their degree of involvement in the production process, can be divided into: suppliers of standard components; suppliers of co-designed components; and suppliers of parts entirely designed by the buyer. In this phase, the types and characteristics of the components to be manufactured are disclosed. Then the production costs will be estimated and also the possible ways to rationalize the components identified in the previous phases. Between the design and production phases, sourcing decisions will be made, i.e., choosing which parts must be made externally and which suppliers, and choices relating to internal or external production. Parallel to the internal or external manufacturing of the components, it will be necessary to prepare an assembly phase, followed by finishing, packaging and shipping operations.

The structure and current activities of the Italian branch are inevitably linked to the concept of innovation, which has played, and still plays, a fundamental role in Alpha Optics' strategy. In fact, as claimed by the Innovation Manager:

> [...] innovation has been the engine of Alpha Optics Italy's growth, more than other branches abroad, because Alpha Optics among the large multinationals in the ophthalmic sector is the youngest on the Italian market "since in the 1980s" [...] There were already those who are our main competitors. Therefore, of the big companies, Alpha Optics was the last to approach the Italian market and the one that allowed it to grow a lot and to become, after all, the second company on the market [...] and to reach the results of our main competitors [...] Was to innovate in the product and especially in the services.

In general, innovation appears to be, especially for the vision care division, the one of which Alpha Optics Italy is part, a strategic factor that has made it possible to develop not only products increasingly customized to the needs of end consumers but also to optimize the production and distribution processes within the company as well as providing innovative services or implementing existing ones for the channel customers of Alpha Optics, the optical centers.

This has favored not only the dimensional growth of the company but also filled the gap with competitors who have been present in the sector for some time and with very solid market shares.

Based on this consideration, it can then be seen how the concept of innovation seems to be an innate characteristic of some companies: a pick that allows them to overcome the barriers to entry in a sector, an antibody that allows them to resist competition and pressure from internal players and a highly differentiating factor:

> [...] When Alpha Optics started selling its products on the Italian market, it had no products [...] without the anti-reflective treatment. It was something unthinkable in the 1980s when [...] the big problem of surface treatments was that of resistance, of seal. The Alpha Optics product had a much higher quality than its competitors. It was also the reason why there was no untreated lens for Alpha Optics [...]. Product innovation has been a great engine of development

[...] because Alpha Optics has always had its own philosophy, even against the tide [...] with proposals that could have been little understood at that time.

And again, when the natural predisposition for innovation is combined with the use of the most modern technologies, it will end up being ahead of the times:

Speaking instead of services, the big innovation that brought Alpha Optics to Italy in the early 2000s [...] was to provide the remote shaping service which, in some way, is a principle from which the so-called Industry 4.0 is inspired. and that we introduced when there was no talk of Industry 4.0. In fact, through the joint use of software and hardware it was possible to dematerialize a process that, otherwise, took place manually within the optical center. [...] traditionally, the lens company produced the lenses and sent them to the optician [...] who in his assembly workshop went to shape them on the shape and size of the frame chosen by the end customer. [...] Alpha Optics has implemented an online ordering system thanks to software, where the optician can enter the type of lenses, the treatments, the prescription, all the characteristics that the lens must have, and a hardware, which we call a tracker, which detects the shape of the frame and sends it inside the data packet. Alpha Optics, in addition to producing the lens according to the characteristics that are requested, goes to shape the lens according to the final shape, without moving the frames.

The early implementation of some of the Industry 4.0 technologies, in the case of Alpha Optics, has generated an innovation process that has led to a total reconfiguration of the company's value proposition as suggested in literature [18,23,24].

This service, now offered by almost all companies in the sector, was implemented by Alpha Optics in all its parts: from the online ordering software, to the tracker up to the acquisition of numerically controlled industrial machines that allowed it 100% control over the size of the lens based on data provided by the optical center.

Once the centrality of the concept of innovation in Alpha Optics was understood, it was possible to investigate how the innovation processes in the company start. As it has been from the semi-structured interview, Alpha Optics does not seem to have an independent research and development division. It is possible to assert that the role of the Italian branch is to introduce into the market what is created by the parent company. Therefore, it seems to be as Cooke [43] describes modern companies, that there is geographical separation of management and innovation (including R&D) from production engineering and, finally, routine assembly. However, there is a certain freedom on the part of the branch to introduce innovative ideas such as, for example, with regard to everything related to colored lenses:

[...] We are one of the leading countries for everything related to colored lenses, to the point where the Alpha Optics Italy colored lens collection is different from that [...] of Alpha Optics colored lenses in other countries. To the point that some countries also introduce Alpha Optics Italian colors, which go beyond the normal range.

Furthermore, there is a continuous exchange of information and ideas between members belonging to different branches around the world with the creation of teams that jointly carry out projects on a global level [44].

These considerations allow the understanding of how the innovation processes are so often promoted and introduced by the parent company but that, at the same time, they can be the result of the activity of the branches, which introduce incremental or radical innovations, in order to be more compliant with the needs of the reference market. This is in line with the paradigm of open innovation within an Enterprise 2.0 context, which opens up the classical funnel to encompass flows of technology and ideas within and outside the organization: the duration of creation, recognition and articulation of opportunities can be drastically shortened if ideas come not just from the R&D department [45]. Open innovation

is defined as the use of purposive inflows and outflows of knowledge to accelerate internal innovation, and expand markets for external use of innovation, respectively [46].

*4.3. Product Innovation and Industry 4.0 in Alpha Optics*

Subsequently, the focus of the interview was shifted to product innovations. Among these, we find one of the latest innovations recently launched by Alpha Optics:

> It is an accommodative support lens that is not only addressed to the pre-presbyopic but [...] to all the people who spend most of their time looking at close distances. A need that has been greatly emphasized in recent years with the use of all digital devices.

The Innovation Manager also talked about a radical innovation not yet present in the Italian market and that Alpha Optics is preparing to launch in November 2020. This is another lens:

> [...] A novelty for ophthalmic lenses because it is a lens for the control of myopic progression. It is a lens that Alpha Optics has already introduced in Asian markets, where there is a greater need for this type of lens given the very high incidence of myopia compared to Western countries. It is a lens that controls myopic progression with the aim of reducing its progression [...] but blocking it. It is a product that, indicatively, will be addressed to an age range between 6 and 18 years.

Looking at these two radical innovations, the connection with the trends that influence the demand for eyeglasses in consumers is clear; on the one hand, a product intended for a range of consumers who use digital devices continuously can be found; on the other, a product specifically designed to counter the progression of myopia in young subjects. It is no coincidence that this lens was launched in Asian countries, where the incidence of myopia reaches rates of 97%, and is about to land in Europe, in high-income countries where a positive correlation between high income groups and onset of juvenile myopia has been detected.

Among the product innovations, the most interesting was the one introduced by another project, which sees the direct involvement of one of the enabling technologies of Industry 4.0: additive manufacturing. This project represents not only an innovative way of conceiving the manufacture of eyeglasses, but also a product that, in some way, shifts the consolidated ritual purchase that sees the choice of the frame as the first step and the choice of the lens more suited to the same. It has been asked of the Innovation Manager to investigate the logic behind this project to understand the role of 3D printing:

> [...] all this comes from a thought, that is, Alpha Optics has been offering customized lenses for years, and it means that the lenses are built taking into account personal parameters and in particular the parameters [...] of the specific frame on which lenses will be mounted. Alpha Optics wondered what could have been the next step to further improve the quality of vision [...] and an answer that was given was [...] we need to have more control over the position of the lenses. [...] Then we build the frame based on the position of the lens and based on the anatomical characteristics of the person [...] we customize the size, we customize how the frame fits on the nose, fits on the ears [...]. Right now the only technology that allows us to make each pair of glasses different from the others is 3D printing [...] additive manufacturing.

Additionally, in this case the birth of this project can be interpreted as a way for Alpha Optics to grasp the ongoing trend of the demand for tailor-made eyewear. Unlike other competitors, who seem to make use of artisanal methods for the manufacture of eyewear, however, an additive technology, named additive manufacturing, was chosen for the realization of these models deriving from the project, that is defined as "a process of joining materials to make objects from 3D model data, usually layer upon layer, as opposed to subtractive manufacturing methodologies" [1].

The use of this technology might lead Alpha Optics to consider opportunities for mass customization of its products, local (distributed) production, and no-waste printing process as suggested in literature [20,21], introducing elements of sustainability in its product innovation process and developing sustainable and smart products (SSP) [25,26].

However, for now it seems that for Alpha Optics the use of additive manufacturing seems to be instrumental only to the creation of the product resulting from this project. In fact, one could legitimately be led to imagine the collaboration with a leading company in 3D printing in Europe, called for privacy reasons 3D Manufacturers as a way for Alpha Optics to approach a business that does not belong to it, such as that of optical frames, through a niche product and/or as a way to test or acquire knowledge on additive technology to subsequently start the internal management of the production of frames via 3D printing. However, these are hypotheses which do not seem to fall within the plans of Alpha Optics:

> Interviewer: "If we could look to the future of the project with additive manufacturing techniques [...] could Alpha Optics think about acquiring knowledge of 3D machinery? [...] To manage production internally?
> Innovation Manager: "Certainly from a technological point of view Alpha Optics could do it. Also of skills, the point is that at this moment it is not on the agenda because it is a project that arises from a collaboration [...] between us and 3D Manufacturers, it is a product of both"

In the collaboration with 3D Manufacturers, the sharing of skills between the two companies and other parties involved in the project emerges:

> It is a sharing project between two companies which then dealt with each of its areas of expertise. 3D Manufacturers took care of [...] all the part related to 3D printing but also all the software design and development part and Alpha Optics instead brought its expertise in the field of ophthalmic lenses, product customization [...] but then, to have design skills and also ergonomics ones and everything needed for the eyewear product, we also made use of the collaboration—at first—of a Belgian designer.
> Interviewer: "So, for Alpha Optics, this project is just a way to give extreme prominence to their products [...] or a way to look into a business that, let's say, is naturally connected but in which Alpha Optics does not currently operate, namely that of frames?"
> Innovation Manager: "[...] the goal was another, to take a subsequent step forward in the quality of vision for the consumer [...] and it seemed to us that this could be a way to achieve it. And it's not so much tied to a desire to enter the world of frames."

From the words of the Alpha Optics Innovation Manger, it is clear that the use of additive manufacturing for this project represents the way to introduce "[...] a true innovation because it is a product that did not exist on the market, that is the idea of an eyewear that was born on the visual needs and not on the aesthetic ones" and therefore a way to give greater prominence to the products already marketed by the company.

From the interviewee's knowledge it emerged that Alpha Optics uses five of the nine enabling technologies of Industry 4.0, as summarized in Table 4.

Closely related to the issues of innovation and its use in Industry 4.0 are the implications that this type of business setting has on the human capital of the company. In fact, the setting of the production system of the smart industries, while not canceling the Fordist structure of the latter, is integrated by new technologies, delimiting a new workspace in which the interaction between the physical and virtual world is essential. With Cyber-Physical Production System there is not the disappearance of the human factor but there is a shift towards more conceptual roles and tasks, marking the transition from the traditional figure of the simple worker (blue collar), assigned to physical tasks, to that of a "skilled worker", (white collar) [1,47,48].

**Table 4.** The enabling technologies of Industry 4.0 used by Alpha Optics.

| Enabling Technology | Description |
|---|---|
| Advanced manufacturing solutions | considered as the use of interconnected and rapidly programmable collaborative robots, they have been used by Alpha Optics for several years (late 1990s) in the robotization of lens construction and control processes. |
| Additive manufacturing | That is the use of 3D printers connected to digital development software, turned out to be a technology that Alpha Optics indirectly uses for the creation of a particular type of eyewear, relying on an external partner such as 3D Manufacturer who "*already had expertise in "the area of production of frames with 3D printing [...] but also of all the design and development of software*" suitable for molding the frame. |
| Augmented reality | it is not exploited by the company |
| Simulation | That is, simulation between interconnected machines for the optimization of processes, it is used in the company in the design and development of complex lenses, such as progressive lenses, in which "*[...] the movement of the eyes is simulated to build the design of the lens in a way such that the areas where the vision is clearest are in the points where the eye goes frequently [...] the simulation of the eye movement allows us to [...] modify the design of the lens based on the ocular movement, therefore the simulation software we have been using it for many years*". |
| Horizontal/vertical integration | That is, the integration along the value chain from supplier to consumer, is a fundamental part of Alpha Optics Italy division's activity, which has been using "*[...] software [..] integrated and which contains all the data for managing the order in any part of the process which is perfectly integrated with the remote ordering system [...] one of the [...] strengths of Alpha Optics*". |
| Industrial internet | it is not exploited by the company |
| Cloud | for the management of large amounts of data on open systems, it seems to be present above all for what concerns the management and reception of orders. In general, Alpha Optics's systems "*[...] work with a cloud computing concept*". |
| Cyber security | That is, the safety during the operations in the network and on open systems, is used in the company and is expressed also in limitations on use or access to certain information for security reasons. |
| Big data and analytics | it is not exploited by the company |

This theory is also confirmed in the words of the Innovation Manager:

[...] The employee has had to change their skills over time [...]. We have certainly increased automation because [...] then it also brought benefits to the product [...]. So, people are used in a different way [...] an improvement change [...] in all areas, even in the office work part.

*4.4. Quality and Lean Organization in Alpha Optics*

The last issue addressed during the interview brought out two issues in particular: the adoption of methodologies typical of the so-called lean organization and the presence of system certifications in Alpha Optics. As for the first issue, it has been seen how the lean management culture, despite being born in a sector, the automotive, that is far from that of a company operating in the eyewear sector, has also had an effect on Alpha Optics:

Let's start with the production part. In the local production of Alpha Optics Italy we apply some of the lean techniques: for example, continuous improvement, the 5s method, which is that of the standard organization of work, and then that linked to the maintenance of machinery and above all the involvement of people. This is something that we have been implementing for a few years now: involving people to bring about improvements and that has given us positive results. [...]. The lean organization has brought excellent results in the production department and I must say that the production manager has told us about it over the years because it is something he is very proud of. It was a project that was wanted and that brought us excellent results both in terms of quality and

efficiency but also a lot in the motivation of people, who are directly involved in the improvement process.

As for the management systems certifications, Alpha Optics Italy division is ISO 9001:2015 certified, for all that concerns the requirements for the development of a Quality Management Systems; ISO 14001:2015, for everything concerning Environmental Management Systems; and ISO 45001:2018 for Occupational Health and Safety Management Systems. With particular reference to ISO 9001:2015, the impact that this certification has had on the company, especially in its growth phase, should be noted:

> There is certainly a point on certifications. They involve the continuous updating of procedures—now I speak more of ISO 9001:2015 [...]—which also leads you to optimize processes. The moment when you revise the system, it is a moment, in some way, also for discussion and thinking about how the activities are carried out. I remember that in the 90s when we made the first ISO certification it had a great impact because it brought us to a company that was growing a lot but that perhaps had little culture of leaving a trace of its processes. It made us face everything in a different way: the fact of having to describe a process has already led us to think about things that we didn't care about before.

## 5. Conclusions

The qualitative analysis conducted through a semi-structured interview with the Innovation Manager of Alpha Optics Italy division has the primary objective of understanding the role of enabling technologies of Industry 4.0 in product and process innovation for developing health products in a company operating in the eyewear sector (in the broadest exception considered here). The research question that wanted to be discussed was: *How can the enabling technologies of Industry 4.0 be used to introduce elements of quality and product and process innovation in the eyewear sector?*

The examined company achieved a good position on the market, where there were already many competitors thanks to innovations. This in the first instance allows us to emphasize how innovation can be considered as one of the most important contemporary sources of competitive advantage.

From the study of the research results, it can be asserted that three fundamental pieces of evidence emerged.

The first it is that Alpha Optics is using more than half of the enabling technologies of Industry 4.0, but also that these technologies were adopted in the company before they were defined as such and led back to the concept of the fourth industrial revolution. It can be argued that this could be attributable to three main factors. Among these there is an innate predisposition to innovation, as a consequence of a well-defined corporate culture, and an economic factor, characterized by a high capacity for investment in innovation due to the size of the company. To these two can be added the need, on the part of Alpha Optics Italy division, to penetrate the Italian lens market which, in the 1980s, was populated by companies with a well-established position. The use of new technologies in the innovation of both processes and products has been for Alpha Optics, not only as the only means known by the company to compete but above all as a factor in accelerating the growth of the Italian branch, which, in a few years, has managed to fill the gap with the major giants in the sector.

The second evidence provided by the interview concerns the difficulty of separating and clearly identifying the areas in which one or the other technology is used. Process and product innovation are closely connected and refer to each other, sometimes they are perfectly integrated as in the case of remote shaping. The construction and implementation of a machine capable of tracing the shape of the frame (product innovation) ends up having an exponential impact on the lens ordering and eyewear assembly process.

From this and other examples derives a third evidence: that of the increasing use of new technologies to enhance the services offered to the optical center. In this case, it is a trend that can be found both in the strategy of this multinational company and in other

companies in the sector. Eyewear products, more than others, require the presence of a specialized figure, the optician, who intervenes in the purchasing process. Except for a few flagship stores of some brands, in Italy the game between different players is in the retailer. Implementing services to support the optician's sales practices means making it, on the one hand, more independent (for example, the remote shaping service allows the optician not to have to process the lens himself at his own optical center, or to rely on specialized laboratories, and to be able to devote more time to sales activities) but, at the same time, makes him an ambassador of the provider of this service.

Furthermore, in the first place, from the Alpha Optics case study it was possible to observe how the connection with the trends that influence the demand for eyeglasses is a driving factor for product innovation. Products increasingly adapted to the needs of young people and the use of digital devices seem to be the ones on which the greatest number of innovations are concentrated.

Secondly, the interview highlighted some aspects relating to the impact that the introduction of technologies has on human capital. In particular, it was confirmed that the growing interaction between the physical and the virtual world has led to the emergence, in Alpha Optics, of the so-called Cyber-Physical Production System, a phenomenon characterized by a shift of the human factor towards more conceptual roles and tasks, which marks the transition from the traditional figure of the simple worker to the skilled worker.

Considering these points emerging from Alpha Optics activities and the high use of Industry 4.0 enabling technologies, a starting point for improvement can be defined: the company could think of an even bigger change in its business model, considering in a more relevant way the relationship between innovation and sustainability, and adjusting the innovation strategy by implementing a sustainable business model to reduce their environmental impacts [27].

Through the qualitative survey carried out in this work, it was possible to closely observe the reality of a large multinational such as Alpha Optics. An in-depth study that would equally closely investigate the approach of Small and Medium size enterprises (SMEs) to innovation and the role it plays within the company would therefore be interesting. Furthermore, it would be useful to examine the reasons that push SMEs to certify, given that, of course, the enormous differences in size and financial availability would certainly reveal market dynamics and behaviors that are profoundly different from those that emerged in this analysis.

In conclusion, it can be affirmed that through qualitative research it has been possible to look more closely at the research questions and to approach an answer which, due to the mutability of the context, certainly cannot be defined as definitive. Innovation seems to respond to different logics than those used to "crystallize" the state of the art.

There is still an open debate not only regarding the understanding of which industrial revolution we are experiencing but also on the definition of the intrinsic characteristics of this revolution.

Companies such as Alpha Optics that were probably already 4.0 when 4.0 was not yet named, can be defined as forerunners of a revolution which broke out only recently, due to the universality and intensity of use of these technologies.

The main limitation of this study concerns the fact that a single case study has been taken as reference for this research, which has helped the researcher to go deep into the case analysis, bringing out interesting qualitative results, but it does not allow the results achieved to be generalized to the entire sector. For future research it could be interesting both to deepen the research on the impact of innovative solutions on the company's market results supplementing it with a panel of experts who will assess the conditions for implementing industry 4.0 solutions, presented in the case study, in the eyewear sector; and also consider to develop a quantitative analysis to understand the process and product innovation strategies of a large number of companies belonging to the eyewear sector, and how Industry 4.0 enabling technologies are used to develop their innovative strategies.

**Author Contributions:** The work is the result of the fair collaboration of all the authors. F.M. and G.S. supervised the research process and was responsible of the literature analysis. L.B. collected and curated the data and defined the methodology. F.M. and L.B. discussed the conceptualization and the results interpretation and all three together drew the conclusions. All authors have read and agreed to the published version of the manuscript.

**Funding:** This research received no external funding.

**Informed Consent Statement:** Informed consent was obtained from all subjects involved in the study.

**Data Availability Statement:** Not applicable.

**Conflicts of Interest:** The authors declare no conflict of interest

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
