# Peer review of "Sustainable Process and Product Innovation in the Eyewear Sector: The Role of Industry 4.0 Enabling Technologies"

_sustainability, doi:10.3390/su13010365_

Round 1

Reviewer 1 Report

The subject of the article is very interesting and socially important. The authors discuss interesting issues concerning, for example, the global smart glasses market, ecological trends in the eyewear sector or logistics innovations. The purpose of the work has been defined. The research method and research tool have been selected correctly. In the case of qualitative research and case study analysis, it is worth ensuring a greater margin of discretion for the researcher (partially structured interview).

The authors posed one research question. The research question is interesting and developmental, but the research methods and subjective scope of the research do not allow to answer it. The specifics of the authors' research would be better suited to the question: How the enabling technologies of Industry 4.0 can be used to introduce elements of quality and product and process innovation in the eyewear sector? The question asked in this way does not require a broader view and a large research sample from the researchers, it refers to the example presented in the article, which can be considered good practice in the sector.

The choice of the respondent is correct (one of the companies most committed to innovation). This only confirms that the surveyed company should be treated as a benchmark, not a typical representative of the sector.

I propose two directions of changes: 1) change the goal and research question, 2) deepen research on the impact of innovative solutions on the company's market results and supplement them with a panel of experts who will assess the conditions for implementing Industry 4.0 solutions (presented in the case study) in the sector optical (ocular).

Minor Notes:

- Table 1 presents data about which I do not know which area is related to: world, Europe, EU? It is worth clarifying. To compare data from a wider market to the domestic market, you can use a strategic gap analysis, which compares the trends (course) of processes on a global and local basis.

- An important observation is that the examined company achieved a good position on the market, where there were already many competitors thanks to innovations. This allows us to see innovation as one of the most important contemporary sources of competitive advantage. This was not particularly highlighted in the research results.

- Industry 4.0 does not only mean a group of technologies used in the economy, but above all the universality and intensity of their use. In this context, the claim that one cannot talk about Industry 4.0 because certain technologies were used in individual companies is unjustified.

I recommend that the paper be published after the purpose and research question have been corrected or after the report is supplemented with the results of the research in the panel of experts or by another method that deepens and extends the scope of the research.

Author Response

Dear Editor, dear Reviewers,

Thank you for giving us the opportunity to revise our paper and resubmit it to Sustainability.

We would also thank the Reviewers for their time and effort in reviewing our paper.

We have tried to follow in detail all the relevant comments made by the Reviewers. Below are the specific answers to each comment.

We hope that this new version of our paper would better satisfy your requests.

Best regards,

The Authors.

REVIEWER’S 1 COMMENTS

Comment 1:

The subject of the article is very interesting and socially important. The authors discuss interesting issues concerning, for example, the global smart glasses market, ecological trends in the eyewear sector or logistics innovations. The purpose of the work has been defined. The research method and research tool have been selected correctly. In the case of qualitative research and case study analysis, it is worth ensuring a greater margin of discretion for the researcher (partially structured interview).

The authors posed one research question. The research question is interesting and developmental, but the research methods and subjective scope of the research do not allow to answer it. The specifics of the authors' research would be better suited to the question: How the enabling technologies of Industry 4.0 can be used to introduce elements of quality and product and process innovation in the eyewear sector? The question asked in this way does not require a broader view and a large research sample from the researchers, it refers to the example presented in the article, which can be considered good practice in the sector.

The choice of the respondent is correct (one of the companies most committed to innovation). This only confirms that the surveyed company should be treated as a benchmark, not a typical representative of the sector.

I propose two directions of changes: 1) change the goal and research question, 2) deepen research on the impact of innovative solutions on the company's market results and supplement them with a panel of experts who will assess the conditions for implementing Industry 4.0 solutions (presented in the case study) in the sector optical (ocular).

Response 1:

Thank you very much for considering our paper of value. You gave us a precious suggestion, in fact it is true that changing the research question as you suggested does not require a broader view and a large research sample from the researchers, and this suits well the goal of our work. As for deepening the research on the impact of innovative solutions on the company's market results supplementing it with a panel of experts who will assess the conditions for implementing Industry 4.0 solutions (presented in the case study) in the eyewear sector, this is a good suggestion to develop for further research. We have added it as a suggestion for future studies.

Comment 2:

Table 1 presents data about which I do not know which area is related to: world, Europe, EU? It is worth clarifying. To compare data from a wider market to the domestic market, you can use a strategic gap analysis, which compares the trends (course) of processes on a global and local basis.

Response 2:

Thank you for this important suggestion. We have now specified that the data presented are world data on this sector. Thank you also for the suggestion of the strategic gap analysis, that we will certainly use for future studies.

Comment 3:

An important observation is that the examined company achieved a good position on the market, where there were already many competitors thanks to innovations. This allows us to see innovation as one of the most important contemporary sources of competitive advantage. This was not particularly highlighted in the research results.

Response 3:

Thank you very much for this important evidence. We have decided to better underline this aspect in the conclusion section, where results are discussed and conclusions drawn.

Comment 4:

Industry 4.0 does not only mean a group of technologies used in the economy, but above all the universality and intensity of their use. In this context, the claim that one cannot talk about Industry 4.0 because certain technologies were used in individual companies is unjustified.

Response 4:

This is a right observation. We have changed this part in the conclusion section, following your advice.

Comment 5:

I recommend that the paper be published after the purpose and research question have been corrected or after the report is supplemented with the results of the research in the panel of experts or by another method that deepens and extends the scope of the research.

Response 5:

Thank you. We hope to have satisfied Reviewer 1 in reviewing the paper.

Reviewer 2 Report

The article is a very interesting study on product innovation. I think this study is valuable. If the Authors want to publish this text in "Sustainability", the theoretical background should be developed (to increase scientific soundness) and they should confront their results with former research.

I also recommend:

  • Developing the section methodology (explain what alternative methods could be applied)
  • adding a few sentences what is the novelty of the study
  • Developing references list
  • put the research question of the central place in discussion

Author Response

Dear Editor, dear Reviewers,

Thank you for giving us the opportunity to revise our paper and resubmit it to Sustainability.

We would also thank the Reviewers for their time and effort in reviewing our paper.

We have tried to follow in detail all the relevant comments made by the Reviewers. Below are the specific answers to each comment.

We hope that this new version of our paper would better satisfy your requests.

Best regards,

The Authors.

REVIEWER’S 2 COMMENTS

Comment 1:

The article is a very interesting study on product innovation. I think this study is valuable. If the Authors want to publish this text in "Sustainability", the theoretical background should be developed (to increase scientific soundness) and they should confront their results with former research.

Response 1:

Thank you very much for considering our paper of value. A concise theoretical background has been developed (since Sustainability does not request obligatorily a Literature review section in its guidelines), and results have been discussed considering the background analyzed.

Comment 2:

Developing the section methodology (explain what alternative methods could be applied)

Response 2:

Thank you for this suggestion. We have decided to explain the alternative methods that could be applied to this study in the conclusion section, underling the limitations of the methodology of the single case study. We have also made a reference in the methodology section about this.

Comment 3:

Adding a few sentences what is the novelty of the study

Response 3:

Thank you for this precious comment. The novelty of the study has been explained at the end of the Background section.

Comment 4:

Developing references list

Response 4:

The reference list has been developed, adding 23 new references concerning the theoretical background and the methodology section.

Comment 5:

Put the research question of the central place in discussion.

Response 5:

Thank you for this suggestion. We have now tried to better put the research question on the central place of discussion.

Reviewer 3 Report

The paper describes only the first necessary step for an understanding of the application domain. The authors talk about Porter's value chain but it would be more meaningful to represent the primary processes in BPMN and try to make a critical diagnosis of them. In part something is said in the conclusions but if the paper remained so it would only be good corporate advertising.
Furthermore, does the interaction of the Italian branch with the main one in terms of innovation development use Enterprise 2.0? Apparently the description suggests yes but it would be better to structure the paper in a more scientific way.

Author Response

Dear Editor, dear Reviewers,

Thank you for giving us the opportunity to revise our paper and resubmit it to Sustainability.

We would also thank the Reviewers for their time and effort in reviewing our paper.

We have tried to follow in detail all the relevant comments made by the Reviewers. Below are the specific answers to each comment.

We hope that this new version of our paper would better satisfy your requests.

Best regards,

The Authors.

REVIEWER’S 3 COMMENTS

Comment 1:

The paper describes only the first necessary step for an understanding of the application domain. The authors talk about Porter's value chain but it would be more meaningful to represent the primary processes in BPMN and try to make a critical diagnosis of them. In part something is said in the conclusions but if the paper remained so it would only be good corporate advertising.

Response 1:

Thank you for this suggestion. Always taking the Porter’s value chain as a line, we have graphically represented primary and secondary processes of Alpha Optics in Figure 1 and also the phases involved in the production process of an Alpha Optics eyewear, to be better clear in the result section.

Comment 2:
Furthermore, does the interaction of the Italian branch with the main one in terms of innovation development use Enterprise 2.0? Apparently the description suggests yes but it would be better to structure the paper in a more scientific way.

Response 2:

Thank you for this suggestion. Yes, the model of innovation of Alpha optics is in line with the paradigm of open innovation within an Enterprise 2.0 context, therefore we have underlined this element in the result section.

Reviewer 4 Report

The manuscript analyzes a topic widely discussed in the scientific literature, however, proposing an original key to interpretation: the impact of Industry 4.0 technologies on product quality.

The theoretical context should be enriched and updated with the most recent studies on Industry 4.0 and IoT technologies.

The study is based on a single case study through an interview with an apex manger of the company analyzed. To strengthen this methodological approach, the authors should link this choice with the existing literature to scientifically justify the adoption of the single case and the interview.

In addition, it would be appropriate to find the link between the effects of digitization and the improvement of performance in terms of sustainability, as should be expected from the title itself: Sustainable Process and Product innovation in the Eyewear sector: The Role of Industry 4.0 Enabling Technologies. In fact, the entire manuscript lacks any reference to sustainability.

For this reason, I suggest the authors to revise their manuscript before publication, also taking into account the references to literature indicated in the special issue.

Author Response

Dear Editor, dear Reviewers,

Thank you for giving us the opportunity to revise our paper and resubmit it to Sustainability.

We would also thank the Reviewers for their time and effort in reviewing our paper.

We have tried to follow in detail all the relevant comments made by the Reviewers. Below are the specific answers to each comment.

We hope that this new version of our paper would better satisfy your requests.

Best regards,

The Authors.

REVIEWER’S 4 COMMENTS

Comment 1:

The theoretical context should be enriched and updated with the most recent studies on Industry 4.0 and IoT technologies.

Response 1:

Thank you for this important suggestion. The theoretical background (concise) has been added to the study and discussed in relation to the results obtained (even if Sustainability does not request obligatorily a Literature review section in its guidelines).

Comment 2:

The study is based on a single case study through an interview with an apex manager of the company analyzed. To strengthen this methodological approach, the authors should link this choice with the existing literature to scientifically justify the adoption of the single case and the interview.

Response 2:

Thank you for this suggestion. We have added existing literature that justify the choice of a single case study, also underling the limitation of this methodology as asked also by Reviewer 2.

Comment 3:

In addition, it would be appropriate to find the link between the effects of digitization and the improvement of performance in terms of sustainability, as should be expected from the title itself: Sustainable Process and Product innovation in the Eyewear sector: The Role of Industry 4.0 Enabling Technologies. In fact, the entire manuscript lacks any reference to sustainability.

For this reason, I suggest the authors to revise their manuscript before publication, also taking into account the references to literature indicated in the special issue.

Response 3:

Thank you for your suggestion. In the theoretical background we have discussed the effects of digitization and the improvement of performance in terms of sustainability, taking into account also the references indicated in the special issue.

Round 2

Reviewer 2 Report

Great article, congratulations!

Author Response

Thank you for appreciating our work!

Reviewer 3 Report

the paper now has a more appropriate scientific aspect

Author Response

Thank you for considering our article scientific!

Reviewer 4 Report

Dear authors, after reading the manuscript you submitted and verifying the significant improvements you made, I believe that this latest version is suitable for its publication in this journal.

Author Response

Thank you for considering our paper suitable for publication!

This manuscript is a resubmission of an earlier submission. The following is a list of the peer review reports and author responses from that submission.